# A CMOS-MEMS Pixel Sensor for Thermal Neutron Imaging

**DOI:** 10.3390/mi14050952

**Published:** 2023-04-27

**Authors:** Roberto Mendicino, Gian-Franco Dalla Betta

**Affiliations:** 1Center for Sensing Solutions, Eurac Research, Via A. Volta 13A, 39100 Bolzano, Italy; roberto.mendicino@eurac.edu; 2Department of Industrial Engineering, University of Trento, Via Sommarive 9, 38123 Trento, Italy; 3TIFPA INFN, Via Sommarive 14, 38123 Trento, Italy

**Keywords:** solid-state detectors, front-end, monolithic sensors, microstructured sensors, thermal neutron detectors

## Abstract

A monolithic pixel sensor with high spatial granularity (35 × 40 μm^2^) is presented, aiming at thermal neutron detection and imaging. The device is made using the CMOS SOIPIX technology, with Deep Reactive-Ion Etching post-processing on the backside to obtain high aspect-ratio cavities that will be filled with neutron converters. This is the first monolithic 3D sensor ever reported. Owing to the microstructured backside, a neutron detection efficiency up to 30% can be achieved with a ^10^B converter, as estimated by the Geant4 simulations. Each pixel includes circuitry that allows a large dynamic range and energy discrimination and charge-sharing information between neighboring pixels, with a power dissipation of 10 µW per pixel at 1.8 V power supply. The initial results from the experimental characterization of a first test-chip prototype (array of 25 × 25 pixels) in the laboratory are also reported, dealing with functional tests using alpha particles with energy compatible with the reaction products of neutrons with the converter materials, which validate the device design.

## 1. Introduction

Thermal neutron imaging is a highly useful analytical technique that finds applications across different fields. The contrast in neutron imaging, as in X-ray imaging, depends on the cross section of each material composing the sample. The attenuation of X-rays increases with the atomic number of the elements, whereas the trend for neutrons with respect to material mass is the opposite. As a result, thermal neutrons can penetrate deeply in most metals, so they can be used for non-destructive investigations of bulk metal components. In addition, neutrons are suitable for investigating lightweight materials, such as those containing hydrogen or water [1]. The most important applications of neutron imaging are in nuclear engineering for nuclear reactor monitoring [2], in material science for studying the internal structure of materials [3], in archeology for studying ancient artifacts and sites [4], and in homeland security [5].

For many decades, ^3^He detectors have been the “gold standard” for thermal neutron detection due to their high efficiency and low sensitivity to gamma radiation [6]. However, the end of the Cold War led to a decrease in ^3^He production, which caused a rise in its price and the need for alternative detection solutions. Among such alternatives are gas detectors, scintillators, and solid-state detectors [5]. These detectors are chosen based on their spatial accuracy and active area requirements. Boron-lined proportional detectors, boron trifluoride proportional detectors, and lithium scintillators are the most common alternatives to ^3^He. Each of these options has its own advantages and disadvantages in terms of efficiency, cost, and toxicity.

Solid-state neutron detectors are suitable for applications that require high spatial resolution and small active areas. Microchannel plates (MCPs) with proper doping are a viable option providing a good trade-off between high detection efficiency (up to 70%) and high spatial resolution (15 μm), although discriminating gamma-ray interactions with MCPs is difficult since the induced signals are similar to neutrons [7]. Semiconductor detectors are also used with two possible design approaches. The first one involves using neutron-sensitive semiconductors (such as LiInSe_2_, LiSe, or BN), where most of the charge from the neutron reaction is directly transduced within the sensor itself, resulting in a very high efficiency [8]. This approach is challenging due to issues such as crystal quality and defects that can reduce the signal charge collection. Fabrication technologies for these materials are not well established, although promising results have recently been reported [9]. The second approach involves coating semiconductor (mainly silicon) sensors with thin films of neutron converter materials. ^157^Gd and ^113^Cd are materials with the highest thermal neutron cross sections, but their reaction products are γ-rays, which are challenging to distinguish from the background gamma radiation. Hence, other converters such as ^10^B or ^6^Li, despite having lower cross sections, are preferred because their reaction products are charged particles, which favor γ-ray discrimination. This reduces interference and enhances the accuracy of neutron detection. However, the short range of the reaction products is also a disadvantage, mainly because of reaction product self-absorption in the converter layer. Another issue for coated detectors is the dead layer (caused, e.g., by metal contacts, high doping regions, and surface passivation layers) between the active area of a semiconductor detector and the converter layer. In the case of planar detectors coated with ^10^B and/or ^6^Li films, these problems cause the neutron detection efficiency to be at most a few % [10]. Nevertheless, interesting results in neutron imaging have been reported for coated planar pixel sensors coupled to readout chips from the Medipix family [11], as well as for coated imaging devices such as CCDs [12] and CMOS pixels [13,14].

A significant improvement in detection efficiency can be obtained using three-dimensional structures, which are obtained via MEMS techniques. These devices feature deep cavities on their detector surface, which will be filled with neutron-converting materials. As a result, the effective converter thickness and surface area are decoupled from the ranges of reaction products, resulting in a high neutron absorption probability without affecting the detection of reaction products. In addition, if these cavities are designed correctly, conversion products generated in the opposite directions can be collected, and γ-ray rejection can be enhanced. This method has been pursued by various research groups with remarkable results. For instance, Kansas State University [15] and Lawrence Livermore National Laboratory [16] have attained neutron detection efficiency of nearly 70% and 50%, respectively. A comprehensive review of these detectors can be found in [17]. While the primary objective of these accomplishments is to substitute ^3^He gas-filled proportional neutron counters for homeland security purposes, some of the suggested approaches can be adapted to create pixelated systems that could be utilized in the field of neutron imaging.

In this paper, we present a monolithic pixel sensor for thermal neutron imaging developed in the framework of the INFN DEEP_3D project (Detectors for neutron imaging with Embedded Electronics Produced in 3D technology) [18]. The detector is fabricated using CMOS SOI technology with Deep Reactive-Ion Etching (DRIE) post-processing on the backside, aiming at obtaining high aspect-ratio cavities to be filled with neutron converters. This is the first microstructured CMOS monolithic radiation sensor ever reported, with potential impact on neutron imaging and beyond. In fact, from the technological point of view, DEEP_3D can be considered a first attempt in the direction to combine the advantages of 3D detectors, with their excellent radiation hardness [19], and monolithic design with high performance and compactness [20].

The device concept builds upon the approach that we have previously tested with a custom technology in the INFN HYDE project (HYbrid DEtectors of neutrons) [21,22], aiming at improving both detection efficiency and spatial resolution.

The ultimate, long-term goal of the project is the development of a high-performance detector for the study of archeological samples (e.g., works of art made of heavy materials such as bronze and iron), which requires objects to be analyzed in non-destructive tests with high spatial resolution and relatively low neutron fluxes to avoid material activation. The performance of existing neutron imaging systems is not satisfactory for these applications, mainly because of a too low detection efficiency, often lower than 1%, that calls for using large facilities with bright neutron sources to carry out experiments in a reasonable time frame. In order to advance the state of the art, owing to the much higher efficiency expected with the proposed 3D approach, the DEEP 3D project aims at developing a portable system for in-field neutron tomography. In this application, the DEEP_3D detector will be combined with a small accelerator-material target acting as a neutron source and a moderator layer that can be used for neutron tomography.

The remaining part of this paper is organized as follows: the prior art developed in the HYDE project is reviewed in Section 2; the DEEP_3D device is reported in Section 3, covering technological aspects, device simulations, pixel circuit operation principle, and chip design; Section 4 presents the data acquisition systems; and the experimental results are reported in Section 5. Discussion and outlook follow in Section 6.

## 2. HYDE Detectors

HYDE detectors are microstructured sensors for thermal neutrons made with a simplified, quasi-2D technology: they consist of planar detectors with high aspect-ratio cavities added on the ohmic side (see Figure 1) [21,22]. The substrate is a high-resistivity (HR) p-type material, and the junction side is patterned in n^+^ pixels, which are isolated at the surface by a p-spray layer [23], whereas a uniform p^+^ region is present on the ohmic side. With reference to Figure 1, the working principle of this detector technology can be described as follows: when a neutron interacts with the converter material, depending on the chosen converter, it produces α particles and other ions (**^7^**Li and **^3^**H for **^10^**B or **^6^**Li, respectively) with a given energy [17]. The goal of the silicon sensor is to detect these charged particles (ideally both of them), which will deposit their energy close to the walls of the cavities to generate electron–hole pairs. Holes will be collected at the backside p**^+^** contact, where reverse bias is applied at the chip periphery. Conversely, electrons will move toward the frontside n**^+^** pixels and induce current pulses, which are fed to external amplifiers.

The pixel pitch is 55 μm for compatibility with readout chips of the MEDIPIX/TIMEPIX family [24]. The cavities are etched using DRIE (Deep Reactive-Ion Etching [25]) at the end of the process, using the backside metal as a mask for etching. By doing so, the walls and the bottom of the cavities are not doped: while this reduces the dead layer thickness, it requires the cavities to be passivated in an alternate way. For this purpose, a thin (~50 nm) layer of Al_2_O_3_ is deposited based on Atomic Layer Deposition (ALD). After annealing, the Al_2_O_3_ layer exhibits a negative fixed charge in the order of −10^12^ cm**^−^**^2^, which can effectively suppress surface generation/recombination arising from the residual damage due to DRIE [21,22,26,27]. However, it should be noted that the presence of the negatively charged Al_2_O_3_ layer prevents the silicon regions in between the cavities from being fully depleted. As a result, the electron–hole pairs generated by the reaction products within these regions must move first via diffusion, leading to relatively slow signals in the order of several microseconds. This property was used in [22] to achieve a good **γ**-ray suppression without signal loss through pulse-shape analysis of signals processed with different shaping times.

HYDE devices are designed with a cobweb shape for the layout of the cavities, featuring elementary blocks of the same size for the pixel pitch, i.e., 55 μm (see Figure 2 left). Initially, the sizes of the cavities and of the silicon walls in between them were designed for a boron converter, which was expected to yield the best results in terms of detection efficiency and spatial resolution at a given pixel pitch [28]. Several technological experiments were performed, and they indicated that limiting the depth of DRIE of the cavities to approximately 25 μm would be necessary. This measure aims to prevent mechanical and thermal problems caused by heat generated during the etching process, since the narrow walls around the cavities make it difficult to dissipate this heat.

Preliminary results were obtained from diode structures having ~25 μm deep cavities partially filled with ^10^B using Low-Pressure Chemical Vapor Deposition (LPCVD) at the Lawrence Livermore National Laboratory (LLNL), Livermore, USA (see Figure 2 right). Under exposure to thermal neutrons from a moderated alpha-beryllium source at Politecnico di Milano, Italy, a detection efficiency of ~7% was measured [27], which was lower than expected because of an insufficient thickness of the boron layer in the tested sample (~500 nm).

For HYDE pixel detectors, the need for bump bonding to the readout chip poses significant problems when using a boron converter. In fact, the temperature necessary for the LPCVD of boron (~350 °C) is not compatible with detector assemblies after bump bonding. In principle, the deposition of boron could be made on bare silicon sensors before bump bonding, thereby protecting the pixel side from undesired boron deposition. However, this approach may negatively impact the surface quality, thus hindering the bump bonding process. For these reasons, it was decided to process a few wafers of detectors with a different mask for cobweb cavities, which were redesigned with larger geometries suitable for a ^6^LiF converter. Pixel sensors from these wafers were bump bonded to the TIMEPIX readout chip [24] at ADVACAM Oy (Espoo, Finland). The hybrid detectors were assembled in a FitPix system at the Czech Technical University in Prague. They are currently being tested after filling the cavities with ^6^LiF powder [29].

## 3. DEEP_3D Detectors

### 3.1. Device Description

Porting the HYDE concept to monolithic technology makes bump bonding unnecessary, thus allowing the use of a boron converter. For this purpose, a CMOS process enabling the full depletion of a HR p-type substrate is required. Possible solutions could be CMOS technologies used so far for depleted Monolithic Active Pixel Sensors (MAPS), where the electronic circuits are insulated from the HR substrate via deep p-wells [20]. Our choice was to use the 0.2 μm CMOS single Silicon-on-Insulator (SOI) technology developed by the Japanese High-Energy Accelerator Laboratory, KEK, and produced by the Lapis Semiconductor Co. Ltd. [30].

A sketch of the DEEP_3D device concept in this technology is shown in Figure 3. SOI technologies provide the most effective isolation between the CMOS transistors, made in a thin silicon layer, and the sensor active layer through a buried oxide (BOX) layer. The sensor layer consists of a float-zone, p-type silicon substrate with a nominal thickness of 300 µm and a resistivity of 7 kΩ cm, yielding a full depletion voltage of ~140 V. The isolation between the n^+^-charge collection electrodes at the Si-SiO_2_ interface is ensured by the buried p-well implants acting as p-stops [23], not shown in Figure 3.

The process is free from latch-up, and the cross section for single-event upsets is very small. The thickness of the HR substrate can be tailored from ~500 μm down to ~50 μm. After substrate thinning, a good ohmic contact is obtained by backside implantation, laser annealing, and metal deposition. Single SOI technology has a drawback known as the back-gate effect, which changes the electrical properties of CMOS transistors due to the voltage applied to the device’s backside [30]. In addition, the buildup of trapped charge in the BOX layer due to ionizing radiation can also lead to a similar effect. The back-gate effect is suppressed by the presence of buried wells, biased to ground and underneath the circuits, which, however, are not effective against the charging of the BOX layer. The latter problem could be solved using a double-SOI process [31], but a single SOI technology is deemed appropriate for a proof of concept.

After the CMOS-SOI chip fabrication is completed, the device requires backside post-processing, including etching of the cavities by DRIE, passivation of the cavities with Al_2_O_3_ by ALD, and boron filling. The cobweb layout of the cavities was designed following the results from the Geant4 simulations, as described in the following section.

### 3.2. Geant4 and TCAD Simulations

Monte Carlo simulations were performed using the toolkit Geant4 [32]. One million neutrons at a thermal energy of 0.25 meV were generated with a perpendicular and uniform spatial distribution along the xy-surface of the device. The reaction products (α particles and ions) that released energy in the silicon substrate were counted using a threshold of 50 keV. The ratio between the number of reaction products and the neutrons defined the efficiency. From a methodological point of view, the geometry of the cavities was defined using CAD and imported directly into Geant4 as described in [33]. The engine physical simulator model appropriate for this simulation was QGSP BERT HP, using the neutron cross section contained in the GDML library, and the density of ^10^B was considered by including enriched B at 99% and ^11^B at 1% with a density of 2.46 g/cm^3^. The simulations were carried out with neutrons impinging from both sides of the detector. Figure 4 compares the simulation results for different combination of geometrical sizes of the cavities (in the range from 2.8 to 5 μm) and the walls in between them (in the range from 1.1 to 5 μm). Since boron deposition filled the cavities completely (which had a depth of 25 µm) and only 1 µm of boron was considered at the surface, irradiations from the two sides yielded very similar results, with a slightly better efficiency for the irradiation from the side opposite to the cavities. The importance of using small dimensions for the cavities and the walls was evident, with efficiency values decreasing from ~30% to ~12% as the sizes increased from the smallest ones to the largest ones.

TCAD simulations were performed using Synopsys Sentaurus to demonstrate the effective collection of the charge generated when a secondary particle hits the sensitive area. The simulation domain considered was a single 3D pixel, and α-particle simulations were conducted. The reverse bias was 100 V, i.e., slightly below the depletion voltage of the non-etched portion of the substrate. The depth of the cavities was 25 μm. A worst-case scenario was considered for the Al_2_O_3_ passivation layer with a low negative fixed charge density of −10^12^ cm^−2^ and a very large surface recombination velocity of 10^4^ cm/s to account for a high residual damage from DRIE.

Figure 5 shows the transient current response induced by alpha particles of 1.47 MeV (i.e., the most probable energy for a neutron reaction with ^10^B) entering the active volume in two different positions, as shown in the inset of Figure 5 (upside down), which were chosen as representative of two extreme cases: (1) at the bottom of the cavities (red line), and (2) at the surface on top of the regions in between the cavities (black line). In case 1, the signal is fast with a pronounced current peak because charge carriers are generated close to the edge of the depleted region and mainly move by drift. The charge collection efficiency in this case is 100%. Conversely, in case 2, charge carriers have to initially move by diffusion, so that the current peak is much smaller and lasts for a few microseconds. The charge collection efficiency in this case is ~80%, with the charge loss being due to the considered worst-case recombination effects in the cavities.

### 3.3. Chip Design and Pixel Operation

A micrograph of the DEEP_3D chip is shown in Figure 6. It includes an array of 25 × 25 pixels, each having a size of 35 × 40 µm^2^; the CMOS circuitry for the voltage and current references; and the IO ring with all the analog and digital components along the edges. In addition, a test structure is present to enable testing the pixel functionality independently of its array implementation. It consists of two pixels, one featuring the same front-end circuit used in the pixel array and another one composed of a charge pre-amplifier and a feedback circuit. The total area of the chip is 2.9 × 2.9 mm^2^.

The pixel CMOS circuitry is shown in Figure 7, while the corresponding layout is shown in Figure 8. The sensor output current is fed to a charge pre-amplifier which input branch uses a telescopic cascode topology (transistors M1–M4), with gain enhancement obtained by means of the auxiliary PMOS cascode (transistors M5–M6). This ensures a gain higher than 10^4^, which, considering the values of capacitance of the single pixel and of the feedback capacitors, allows the efficient detection of the collected charge generated by an event. To reduce power consumption and increase the input stage gain, the fixed current for each branch of the circuit and the size of the transistors have been calculated to ensure that all the preamplifier transistors are operating in weak inversion. Two different values of capacitance, i.e., 150 fF and 100 fF, can be selected by an external gain control signal for the feedback circuit of the charge pre-amplifier, thus yielding two levels of gain suitable for the different signals expected from the reaction products of the boron and lithium converters. On-chip MIM capacitors are used to this purpose, which, in this technology, have a specific capacitance of 1.5 fF/µm^2^. The constant current rate circuit (MF and Mfb) [34] is responsible for discharging the feedback capacitor, enabling all the transistors to fit within the limited pixel size. This design accommodates slow current signals from the low electrical-field sensor region, where charges move only by diffusion, at the expense of a loss of linearity for small charge signals.

The remaining parts of the pixel circuit are shown as blocks in Figure 7 since they are based on standard circuits. The output of the charge pre-amplifier is fed to a comparator with hysteresis, which is designed to withstand fluctuations in noise. The comparator converts the analog pulse to a digital signal, the duration of which corresponds to the energy of the detected reaction product. The comparator output signal is fed to two different branches: a digital trigger logic which output signal remains high until reset, and a switch in an auxiliary circuit that, when activated, charges a capacitor with a constant current. By doing so, a time-over-threshold (ToT) technique can be exploited, storing the signal amplitude information on the capacitor used as an analog memory. The capacitor can be fully charged within the time of a few µs, which is tunable depending on the sensor signal. The capacitor is connected to a 3T circuit [35], enabling reading of the corresponding voltage value pixel by pixel. To summarize the signal path in the pixel circuit, the sensor output, which is usually short (a few µs) and has a low intensity of current (in the µA range), becomes a signal of a few millivolts and tens of microseconds after the charge preamplifier. The time-over-threshold of this signal provided by the comparator charges an internal capacitor, which, when charged by using a constant current, stores a voltage proportional to the input charge. The 3T circuit can then measure the value of this voltage when the pixel’s information is required. The power consumption of a single pixel is in the order of 10 µW with 1.8 V power supply.

Additional circuits are required to control the operation of the pixel matrix. These circuits include selection circuits for rows and columns, analog amplifiers to support the 3T pixel circuits, and general trigger logic. The selection of rows and columns is achieved using a shift register composed of a chain of 25 D flip-flops with a reset function. Digital circuitry is responsible for injecting the bit into the first shift register. The general trigger circuit collects information from all the pixel triggers and records the presence of an event by transmitting the information outside the matrix.

DEEP_3D is the first 3D monolithic device, a promising candidate to significantly innovate the field of neutron imaging. Its more direct competitors are based on planar neutron detectors, i.e., the hybrid detector developed by the Czech Technical University in Prague based on the Medipix-2 chip [11,28] and the monolithic SOI sensor developed by KEK [14]. Since a planar detector coated with a neutron converter is involved in both of these cases, the neutron detection efficiency is limited at most to 3–4%, whereas with DEEP_3D, it is estimated to achieve an efficiency up to 30% in the best case. The spatial resolution is directly proportional to the pixel size. Since each pixel in the DEEP_3D device includes a charge pre-amplifier and other circuits that make the chip fast enough and enable successive analysis (e.g., gamma discrimination), its size of 35 × 40 μm^2^ is not the smallest in this panorama, but it is still small enough compared to the Medipix-2 device (55 × 55 μm^2^), which also includes complex circuitry. The KEK device has a smaller pixel size of 17 × 17 μm^2^, due to a simpler circuit topology, and an estimated spatial resolution of ~4 μm, but its efficiency is only 1.5%.

## 4. Data Acquisition System

The data acquisition boards’ general schematic and a photograph of the measurement setup are shown in Figure 9. To begin, the DEEP_3D chip is connected to a PCB board through wire bonding, followed by its installation on a primary board. This main board comprises eight reference current generators, a reference voltage, and links to the FPGA board. For this project, a Digilent Zedboard was utilized as the FPGA board, which is an evaluation and development board utilizing the Xilinx Zynq-7000 All Programmable SoC (AP SoC). The FPGA board, operating at 100 MHz, generates the necessary digital signals for controlling the sensor chip, resulting in highly precise signal timing despite the sensor’s relatively low frame rate. To save power, the maximum frame rate is limited to 0.8 kHz, but it can be doubled by increasing the amplifier current. The analog signal is acquired using the FPGA’s inner ADC with a resolution of 12 bits and stored in a FIFO register before being immediately transferred to the FPGA’s RAM via Direct Memory Access (DMA), which optimizes data transfer to avoid dead times. The chip trigger event occurs only when the incident particle releases enough charge to exceed the fixed trigger level.

The internal microprocessor, which is programmed in C language, handles data transfer to the PC via a serial port. The acquired data are then processed using the Matlab software. To minimize power consumption, the IO ring amplifiers are switched off after acquisition until the next trigger event.

## 5. Results

Before proceeding to device post-processing, the chip functionality was tested using a ^241^Am source, which decays 100% via α transitions to ^237^Np. The source available in our laboratory has a low nominal activity of 3 kBq. The α energy’s primary peak is 5.5 MeV, providing an accurate emulation of the reaction products resulting from neutrons colliding with the converter materials.

The initial tests were conducted on the test structure with the pre-amplifier and its feedback circuit. The output signal from the pre-amplifier was further amplified by a buffer stage and acquired by an oscilloscope. Given the small pixel size of only 35 × 40 μm^2^ and the limited activity of the source, the event probability was relatively low, with a rate of 1 event every ~12 min. Figure 10 displays the typical response to an α particle: both the signal amplitude (~0.8 V) and the decay time (~300 μs) are in good agreement with values predicted by the simulations and can be utilized for feedback circuitry tuning.

For the measurements on pixel arrays, the following procedure was used:Acquisition of dark signals: Prior to placing the α source on the sensor, measurements were taken in dark conditions. The FPGA firmware allowed acquisition even without a trigger event from the chip. To obtain an average value for the dark signals, the mean of individual pixels was computed, which also evaluated the uniformity of the matrix. The distribution of the mean pixel values was then calculated and fitted to a Gaussian curve. The distribution was found to be narrow, with a mean value of µ = 0.90 V and a standard deviation of σ = 0.03 V.Removal of defective pixels: any pixels that could trigger incorrect detection events were identified and removed from future analyses by setting them to zero.Acquisition of alpha particle events: real detection events were acquired.Removal of dark signals and data analysis: the acquired data were adjusted by subtracting the dark signals to eliminate any offset, and the data were analyzed accordingly.

### 5.1. Irradiation from the Frontside (CMOS Side)

To analyze the behavior of the front-end electronics when signals are fast, frontside irradiation was performed. In this case, the source was placed approximately 2 cm from the sensor surface.

Since substrate depletion begins at the n^+^/p^−^ junction near the buried oxide, the motion of generated charge occurs primarily by drift, leading to fast signals. However, prior to reaching the sensing component, α particles must pass through several layers (e.g., passivation, metals, dielectrics, and electronics), thus causing a non-negligible energy loss. The reverse bias was set to −75 V, resulting in partial depletion of the high-resistivity substrate.

Figure 11a shows the map of signal acquisition in the dark, while Figure 11b shows the acquisition of a single α-particle event obtained by subtracting the acquired data from the dark acquisition. In the case of frontside irradiation, charge-sharing effects between neighboring pixels are seldom observed due to the fast charge collection (the charge cloud does not have sufficient time to diffuse far from the generation point) and the lower amount of charge generated after energy loss in the electronic layers.

To test the entire matrix, the acquisition procedure was carried out for a relatively long period of time (~36 h) due to the low activity of the source. The resulting hit map is shown in Figure 11c. Most unmasked pixels have recorded at least one event.

### 5.2. Irradiation from the Backside (Sensor Side)

When irradiating from the backside, the reverse bias was increased to −200 V so that the 300 µm thick high-resistivity substrate could be fully depleted. To expose the backside to α particles, a hole of 2.5 mm diameter was drilled on the PCB and the sensor was placed on top of it, while the radiation source was placed underneath at a distance of approximately 2 mm from the board. It is worth noting that the tested sensor has a layer of metal deposited on the backside that shields it from ambient light.

As the neutron converter was placed on the backside, more events were acquired compared to the frontside case, for a total of ~40 h. Figure 12 leftshows the hit map, where, again, most of the pixels are operational, and only one row appears insensitive, possibly due to a defect on this specific chip.

Because of the inaccurate technique employed to produce the hole on the PCB, the matrix was not centered perfectly over the hole, and this is reflected in the hit distribution along the matrix. It is worth noting that in the case of backside irradiation, charge sharing between pixels is clearly present, as also shown in Figure 12 right for a single α-particle event. This feature is particularly important for particle discrimination (e.g., for γ-ray suppression), as already demonstrated in previous studies [36].

## 6. Discussion and Outlook

The initial tests performed in the laboratory at the University of Trento demonstrated that the chip works correctly and in accordance with the design specifications. The power consumption is ~10 μW per pixel at 1.8 V power supply, well matching the design value. The theoretical gain of the charge preamplifier is inversely proportional to the feedback capacitance, and using both feedback capacitors in parallel gives a total capacitance of 250 fF and a gain of 4 V/pC. Upon exposure to alpha particles of 5.5 MeV, the measured output signal amplitude is ~0.8 V, corresponding to a gain of ~3.3 V/pC, which is in good agreement with the expected gain while taking into account the uncertainties due to parasitic capacitances and the energy loss of α particles in air and in the dead layers. Additionally, the decay time of the preamplifier output signal (~300 μs) is consistent with the constant current used in the reset circuit. 

The sensor breakdown voltage is larger than 200 V, and it allows a wide operational margin beyond the full depletion voltage. The signals induced by α particles were acquired for a relatively long exposure time (~40 h), thus confirming the correct functionality of the entire pixel front-end circuit and of the external readout circuit and trigger logic. In the case of frontside irradiation, individual pixels were mostly activated, which was consistent with the limited diffusion of the charge generated close to the pixel junction. On the contrary, charge-sharing events were observed in the case of backside irradiation, as expected from the lateral diffusion of the charge cloud (>5 μm) as it drifted through the entire thickness of the substrate.

Due the low activity of the available α source, it would have taken several weeks to accumulate significant statistics, making it impossible to build an energy spectrum or capture an image. Despite this limitation, the measurement results are encouraging and suggest it is possible to proceed with the backside post-processing that is necessary to make the device sensitive to neutrons. 

For the etching of the cavities using DRIE and their passivation, we are collaborating with the Leibniz Institute of Photonic Technology (Jena, Germany), where preliminary tests have been carried out on bare silicon samples (individual dice having a size comparable to the DEEP_3D chip). For this purpose, a mask with a cobweb layout of 55 μm pitch is used, and etching tests have been performed. As an example, Figure 13 left shows a scanning electron microscopic image of the cavities after the DRIE step: the etching depth (~18 μm) is close to the target value (~25 μm), but the width of the cavities is larger than the nominal value, leading to thinning of the separation walls. Further tests are under way to optimize the etching recipe before applying it to real device samples.

For boron deposition, besides relying on external collaborations using Chemical Vapor Deposition or Atomic Layer Deposition, we are pursuing an alternative approach similar to that used for ^6^LiF deposition, which consists of dispersing nanoparticles of ^10^B_4_C (pure boron at the nanoparticle scale has a very high risk of being flammable) in a non-polar liquid that is deposited on the etched surface and fills the cavities by capillarity. After evaporation, only the ^10^B_4_C grains will remain. With multiple depositions, it should be possible to fill the cavities. Preliminary tests have been conducted with encouraging results using alumina nanoparticles instead of boron carbide (see Figure 13 right).

## Figures and Tables

**Figure 1 micromachines-14-00952-f001:**
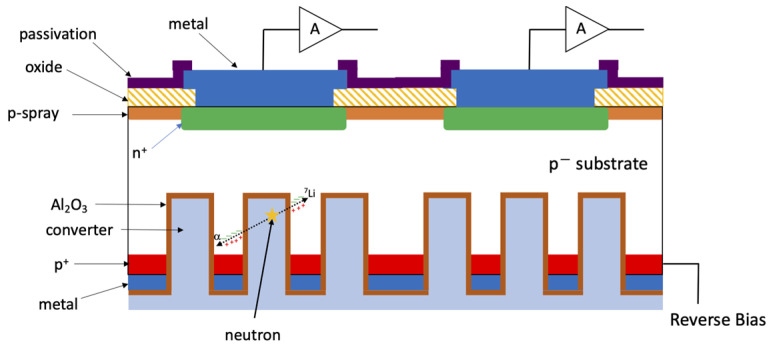
Schematic cross section of a HYDE2 detector (not to scale).

**Figure 2 micromachines-14-00952-f002:**
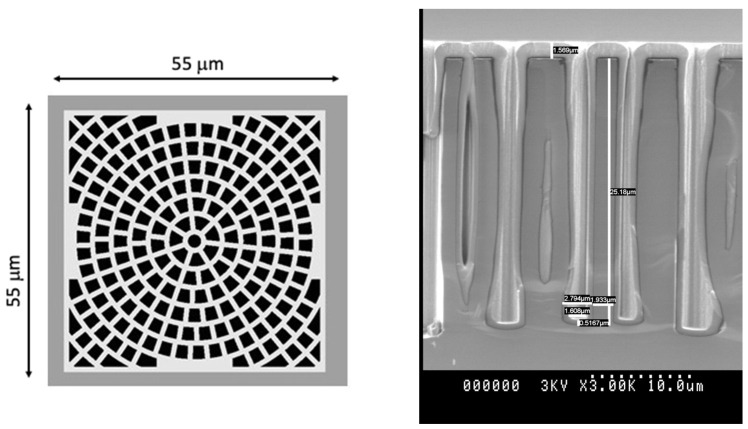
(**Left**) Layout of cavity cobweb (version for ^10^B converter) in a pixel of HYDE detectors. (**right**) Scanning electron microscopic graph of cavities partially filled with ^10^B using LPCVD at LLNL.

**Figure 3 micromachines-14-00952-f003:**
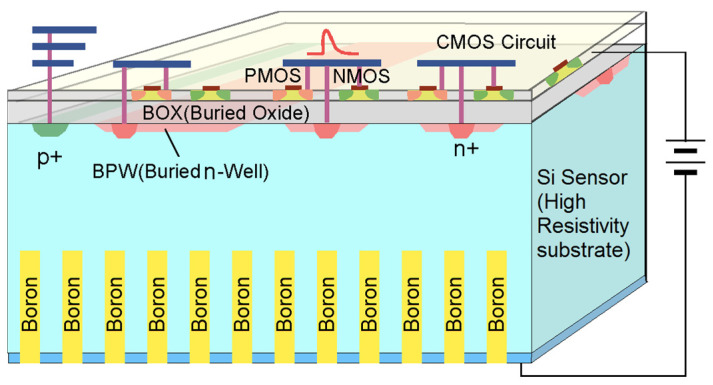
Sketch of the DEEP_3D device based on CMOS SOI technology (not to scale).

**Figure 4 micromachines-14-00952-f004:**
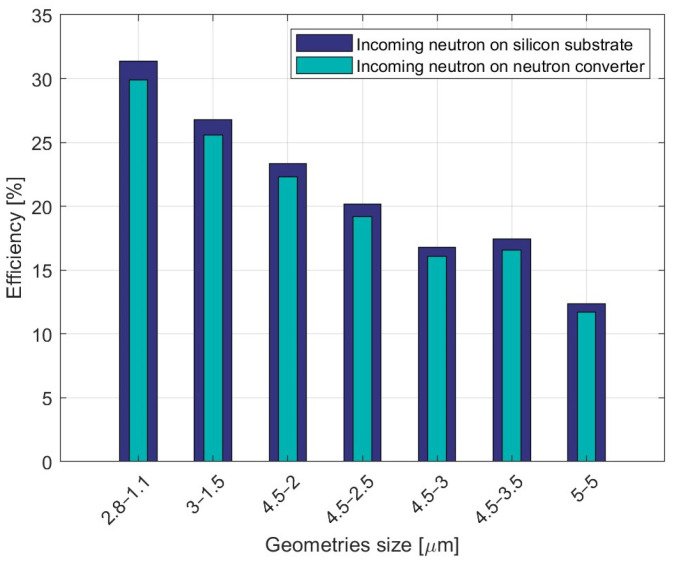
Geant4 simulation of thermal neutron detection efficiency in HYDE 2 devices of different geometrical sizes (the widths of the cavities and of the silicon walls in between them are given in micrometers), with a 50 keV energy threshold.

**Figure 5 micromachines-14-00952-f005:**
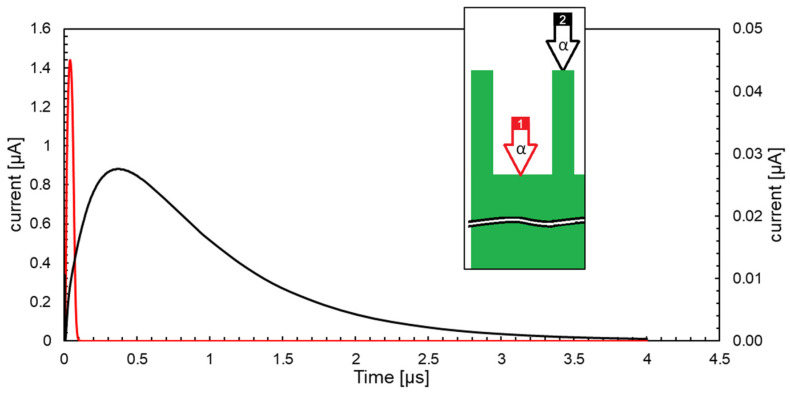
TCAD simulation of output current signal transients for alpha particles hitting position 1, located at the bottom of the cavities (red line, left current axis), and position 2, at the surface on top of the regions in between the cavities (black line, right current axis).

**Figure 6 micromachines-14-00952-f006:**
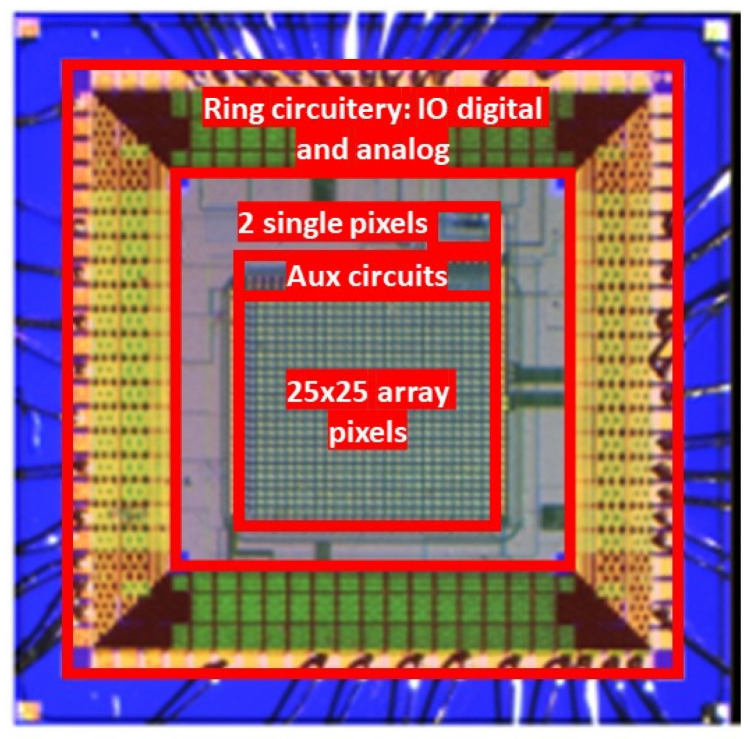
Photograph of the DEEP_3D chip, with different blocks highlighted.

**Figure 7 micromachines-14-00952-f007:**
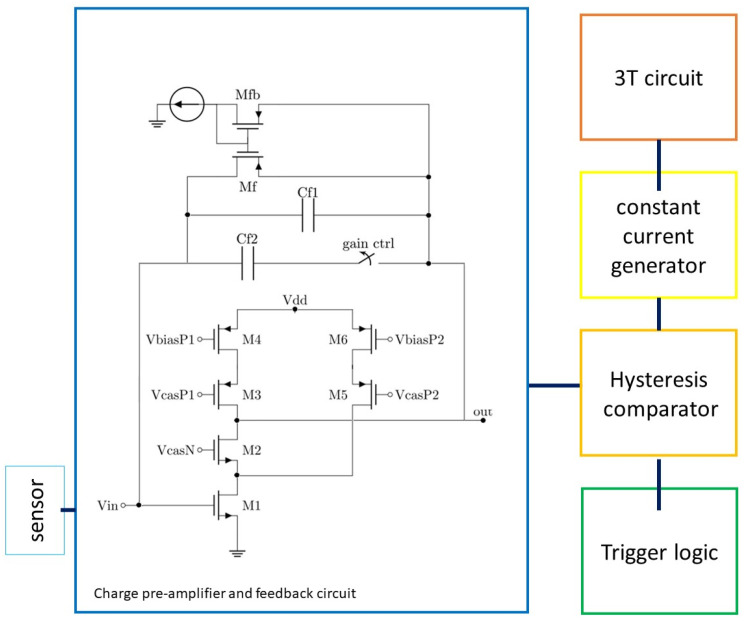
Schematic diagram of the pixel circuit.

**Figure 8 micromachines-14-00952-f008:**
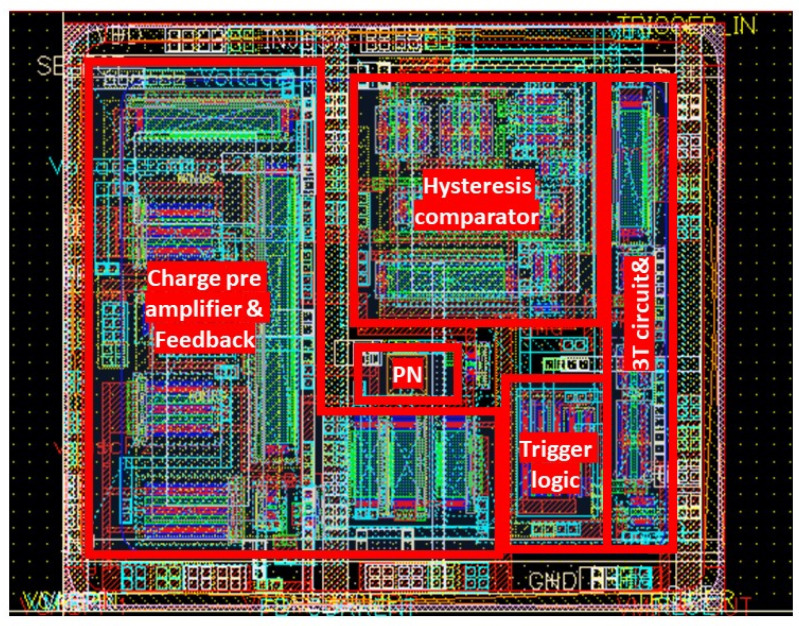
Pixel layout with a size of 35 × 40 µm^2^ with different blocks highlighted. The feedback capacitors, which cover more than 80% of the pixel area and are placed on the entire pixel surface, are not shown.

**Figure 9 micromachines-14-00952-f009:**
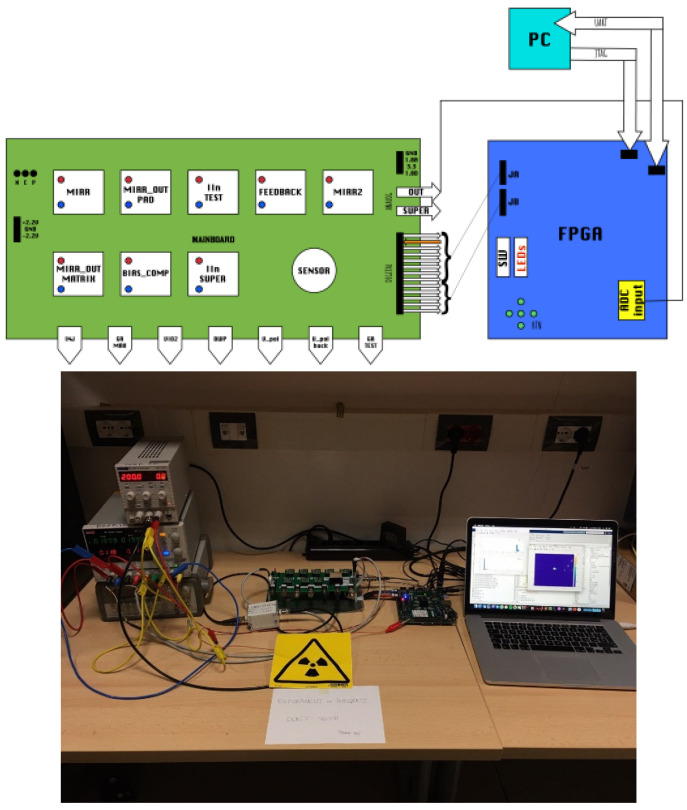
(**Top**) Block diagram of the data acquisition system and (**bottom**) photograph of the measurement setup in the laboratory.

**Figure 10 micromachines-14-00952-f010:**
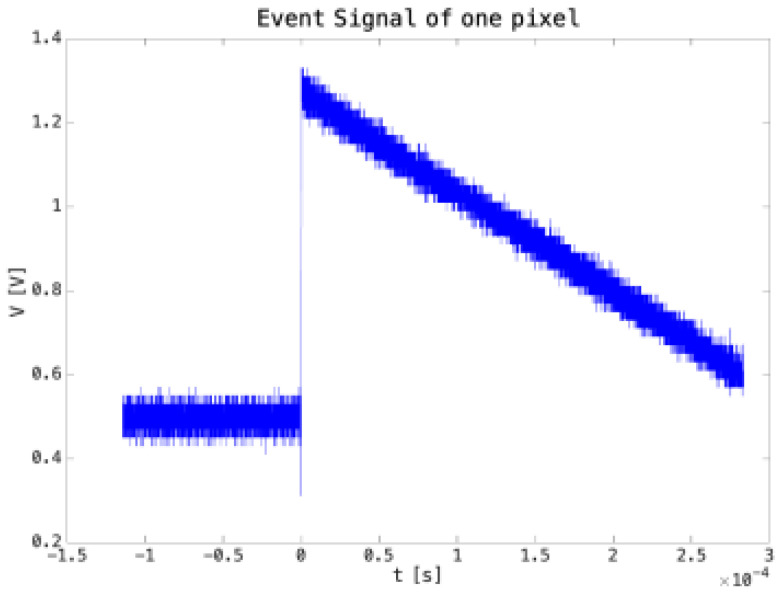
Output signal from the charge pre-amplifier output in response to an α particle from a ^241^Am source. The source is placed at ~1 cm from the CMOS side of the sensor, and both feedback capacitors are selected to manage all the charge released from the ^241^Am source.

**Figure 11 micromachines-14-00952-f011:**

Hit maps for frontside irradiation: (**a**) acquisition of dark signals; (**b**) acquisition of single α event; and (**c**) 36 h acquisition with at least one event in most unmasked pixels.

**Figure 12 micromachines-14-00952-f012:**
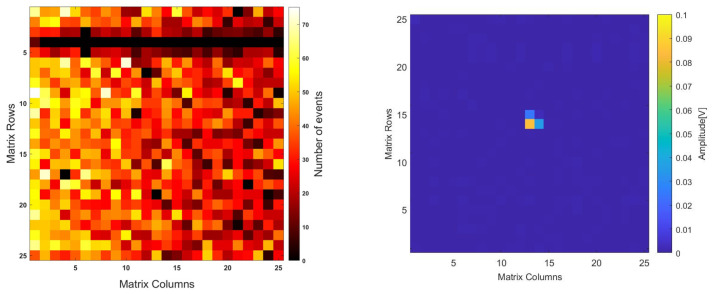
Hit maps for backside irradiation: (**left**) 40 h acquisition, and (**right**) detail of single α event with charge sharing among three adjacent pixels.

**Figure 13 micromachines-14-00952-f013:**
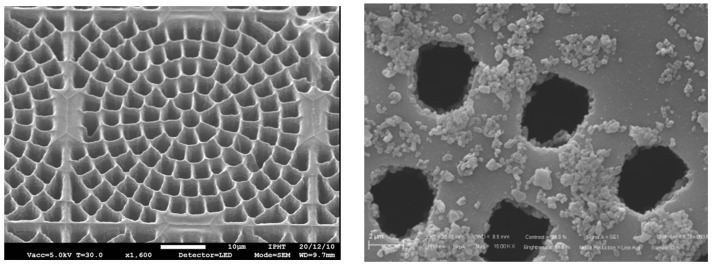
Scanning electron microscopic pictures of (**Left**) cobweb cavities etched by DRIE at Leibniz-IPHT, and (**Right**) deposition test of alumina nanoparticles at the University of Trento.

## Data Availability

Experimental data are available upon request.

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
