# Peer review of "A CMOS-MEMS Pixel Sensor for Thermal Neutron Imaging"

_micromachines, 2023, doi:10.3390/mi14050952_

Round 1
Reviewer 1 Report
The manuscript presented a monolithic pixel sensor with high spatial granularity to detect thermal neutron. And, discussed the sensor design and technological issues, along with simulation results. This study has some guiding significance the design of the monolithic pixel sensor. However, it is obvious theory that the increase in width and length result in a larger noise and longer drop time. Besides this, the model analysis in this paper is not suitably accurate.
1. In the chapter 3, line 216, “Two different values of capacitance, i.e., 150 pF and 100 pF,”. What structure is used of these two capacitors? Whether these two capacitors are inside or outside the chip? The area of 100pF capacitance is too large, which can’t use inside the chip?
2. In the chapter 3, line 215, “the transistors have been sized to operate in 215 the sub-threshold regime”. Please indicate how to works in the sub-threshold region?
3. Please indicate the difference between sensor output current and the 3T pixel output?
4. What is the area of the pixel layout in Figure 7?
5. For a better express the innovations of the article, it is recommended to add comparison (detection efficiency and the spatial resolution) with other studies of the same type at the end of chapter 3.
Author Response
We would like to thank the reviewer for the valuable comments that allowed to improve the paper. Responses can be found in the attachment. All changes made in the revised manuscript are marked up using the “Track
Changes” function in the MS Word file.

Reviewer 2 Report
Paper would be improved by a rewrite that provides a clear motivation based on description of application, that highlights innovation, and that quantifies how performance compares to specification;
Paper is uneven with details of process, but with lack of details of device operation.
Abstract:
· Abstract should highlight innovation.
· Replace “Sensor design and technological issues are discussed, along with simulation results” with key differentiating details.
· Quantify key performance values.
Introduction
· Introduction would be improved if it included a description of application: source, target, and detector arrangement; description of example application and outcome of use.
· Pg 1 Lines 21-23: Clarify why decreasing absorption behavior with mass correlates with imaging lower mass materials; I can guess this is because thinner layers can be measured, but I should not have to guess.
· Pg 2 Lines 85-88: Give look-ahead overview of the advances in the technology in this paper – what is differentiating? Then mention that the details will be described in the next session.
· Pg 2 after Line 88: It would be useful to describe the outline of the rest of the paper – process description of HYDE detector as prior art, description of process and device of DEEP_3D detector as basis of this paper.
HYDE detectors:
· Pg 3 Lines 109-110: Figure 1 should show the location of the neutron-sensitive layer. It is also unclear the extent of the Al2O3 coverage in the wells.
· Pg 3 Lines 104-106: Need simple description of device operation: location of neutron absorption; biasing of device; path for charge collection.
DEEP_3D detectors
· Pg 4 Line 143: Contrast monolithic SOI device structure to HYDE detector prior art. Move Figure 3 reference to here, ahead of details.
· Pg 4 after Line 173: Paper still lacks a device simulation describing the neutron detection: biasing; neutron absorption event; charge transport; readout.
Discussion:
· Pg 10 Line 346: Quantify the specification and performance measured – provide table.
· Pg 10 Lines 355 – 373: This material appears preliminary and disconnected from this paper. It should be considered for a future paper.
Author Response

(The authors gave the same response as above.)

Round 2
Reviewer 2 Report
Revisions make for an improved paper by providing more details to put work in perspective. There are still some gaps — for instance, tomography setup in application, both spatially and temporally. On the other, editing could could improve the flow of the paper.
Figure of structure and explanation of detection physics is significant improvement.